# Significant Improvement of Thermal Conductivity of Polyamide 6/Boron Nitride Composites by Adding a Small Amount of Stearic Acid

**DOI:** 10.3390/polym15081887

**Published:** 2023-04-14

**Authors:** Hui Fang, Guifeng Li, Kai Wang, Fangjuan Wu

**Affiliations:** 1College of Materials Science and Engineering, Fujian University of Technology, Fuzhou 350118, China; 2Key Laboratory of Polymer Materials and Products of Universities in Fujian, Fujian University of Technology, Fuzhou 350118, China; 3Fujian Provincial Key Laboratory of Advanced Materials Processing and Application, Fujian University of Technology, Fuzhou 350118, China

**Keywords:** thermal conductivity, polyamide, boron nitride, stearic acid

## Abstract

This study investigates the effect of adding stearic acid (SA) on the thermal conductivity of polyamide 6 (PA6)/boron nitride (BN) composites. The composites were prepared by melt blending, and the mass ratio of PA6 to BN was fixed at 50:50. The results show that when the SA content is less than 5 phr, some SA is distributed at the interface between BN sheets and PA6, which improves the interface adhesion of the two phases. This improves the force transfer from the matrix to BN sheets, promoting the exfoliation and dispersion of BN sheets. However, when the SA content was greater than 5 phr, SA tends to aggregate and form separate domains rather than being dispersed at the interface between PA6 and BN. Additionally, the well-dispersed BN sheets act as a heterogeneous nucleation agent, significantly improving the crystallinity of the PA6 matrix. The combination of good interface adhesion, excellent orientation, and high crystallinity of the matrix leads to efficient phonon propagation, resulting in a significant improvement in the thermal conductivity of the composite. The highest thermal conductivity of the composite is achieved when the SA content is 5 phr, which is 3.59 W m^−1^ K^−1^. The utilization of a composite material consisting of 5phr SA as the thermal interface material displays the highest thermal conductivity, and the composite also demonstrates satisfactory mechanical properties. This study proposes a promising strategy for the preparation of composites with high thermal conductivity.

## 1. Introduction

Thermal conductivity composites have garnered significant interest in modern electronic devices, LED packaging, and other related fields [1]. However, the polymer matrix’s low thermal conductivity (<0.5 W m^−1^ K^−1^) has limited its potential applications. One feasible solution is adding high thermal conductivity fillers to the polymer. The resulting thermal conductivity composites exhibit high thermal conductivity, low dielectric loss, light weight, and ease of processing.

Polyamide 6 (PA6) is increasingly utilized in various industrial applications, such as aerospace, automotive, and electronics, due to its corrosion resistance, high specific strength, and good formability [2]. Unfortunately, the thermal conductivity of PA6 is only 0.3 W m^−1^ K^−1^ [3], which limits its application in thermal conductive electronic components. Preparing filled PA6-based composites is a beneficial approach to improving the thermal conductivity of PA6. The mainstream thermally conductive fillers currently include aluminum oxide [4,5], carbon nanotubes [6,7], graphene oxide [8,9,10], and hexagonal boron nitride (BN) [11,12,13,14].

BN is a two-dimensional material that has attracted attention for its high in-plane thermal conductivity of 600 W m^−1^ K^−1^ [15]. This property makes it a promising candidate for use as a filler in composites for thermal management, as it can effectively transfer heat while maintaining exceptional chemical and thermal stability at high temperatures [16]. Additionally, BN’s excellent electrical insulation properties, characterized by low dielectric loss and high resistivity, make it an ideal candidate for composites that require both heat conduction and insulation [17,18]. Phonon transmission is the primary heat conduction mechanism in these composites. To significantly enhance the thermal conductivity of these composites, it is necessary to improve the dispersion of BN within the matrix and strengthen the interface adhesion between BN and the matrix. BN’s unique combination of thermal and electrical properties makes it a promising material for a wide range of applications in the fields of electronics, aerospace, and energy.

However, due to BN’s high chemical inertness, meeting the aforementioned requirements is a challenge. To address this, Li et al. [19] utilized solution blending to prepare polyvinylidene fluoride/boron nitride nanosheet (PVDF/BNNS) composites. By subjecting PVDF to appropriate water-bath heating, it improved the dispersion of BNNS and enhanced the fluidity of PVDF. The resulting composite, with a loading of 3.8 wt% BNNS, exhibited a thermal conductivity of 0.49 W m^−1^ K^−1^, which was 3.5 times that of a pure PVDF film. Similarly, Yang et al. [20] developed natural rubber/BN (NR/BN) composites by modifying BN with poly (dopamine) (PDA) and γ-methacryloxypropyl trimethoxy silane to improve the interface bonding between BN and NR. With 51.3 wt% modified BN fillers, the composite displayed a thermal conductivity of 0.39 W m^−1^ K^−1^, which was 3.9 times that of pure NR. Our group [11] prepared isocyanate-functionalized BN (f-BN), dispersed it in caprolactam (CL) solution, and prepared PA6/BN composites through in situ polymerization. Since PA6 was in-situ-grafted on BN sheets during polymerization, BN and PA6 present good interface bonding. The thermal conductivity of PA6/f-BN composites at 5 wt% of f-BN loading was 66% higher than that of pure PA6.

Furthermore, due to the in-plane thermal conductivity of BN being 20 times higher than its through-plane thermal conductivity [21], constructing a three-dimensional BN (3D-BN) network has been proven to significantly enhance the thermal conductivity of polymer/BN composites by enabling phonon transfer along the BN plane. Various techniques have been utilized to fabricate 3D-BN structures, including organic/inorganic templating [22,23,24,25], foaming [26,27], 3D printing [28,29], electric alignment [30,31], magnetic alignment [32,33], and ice templating [16,34,35,36]. For instance, Khakbaz et al. [28] successfully produced thermoplastic polyurethane/BN (PU/BN) composites through 3D printing, resulting in a 74% increase in thermal conductivity at a loading of 20 wt% BN compared with unmodified PU. Han et al. [30] prepared silicone rubber/BN composites using the electric-field-assisted curing technique. Under the AC electric field (50 Hz) of 11.0 kV/mm, the thermal conductivity with 20 vol% loading of BN was enhanced 250% higher than that of the composite prepared without an electric field. In our group’s work [12], we fabricated 3D-BN scaffolds using the ice-templating method and impregnated them with caprolactone monomers, then polymerized them via microwave-assisted techniques to produce polycaprolactone/3D-BN (PCL/3D-BN) composites. The maximum thermal conductivity achieved was 1.42 W m^−1^ K^−1^ at 25.6 wt% BN loading, which was 7.1 times higher than that of pure PCL.

The method mentioned above can dramatically promote the thermal conductivity of the composites. However, direct melt blending is usually the preferred process for the large-scale preparation of thermally conductive materials [37,38,39,40,41,42,43]. Zhang et al. [37] prepared acrylonitrile butadiene styrene copolymer/BN (ABS/BN) composites by melt blending. During the preparation, a small amount of a hyperbranched polymer was added to reduce the viscosity of ABS. With the loading of 60 wt% BN, the thermal conductivity of the composite increased to 1.12 W m^−1^ K^−1^. Wang et al. [38] fabricated PA6/BN composites by the same method. As the BN content was 50 wt%, the thermal conductivity of the composite could reach 0.93 W m^−1^ K^−1^. Generally, when a thermally conductive composite was prepared by the melt-blending method, a large amount of BN fillers was usually added to significantly enhance the thermal conductivity.

In most of the reported work with other methods, although much more BN fillers were applied in the melt-blending process, the thermal conductivity of the composite was still less than 2.0 W m^−1^ K^−1^, which was not enough in some applicants. In this work, PA6/BN composites were prepared by melt blending. By adding a small amount of stearic acid (SA), the thermal conductivity of the composite was dramatically promoted, and the influence of SA on the improvement of the thermal conductivity was investigated.

## 2. Experimental

### 2.1. Materials

PA6 (IMNC101) was purchased from Kuibyshev Azot Engineering Plastics (Shanghai) Co., Ltd. BN microsheets used in this work have a diameter of 3~5 μm, surface area of 4~7 m^2^ g^−1^, and purity of >99% (Qinhuangdao Eno High-Tech Material Development Co., Ltd., Qinhuangdao, China). Pentaerythritol tetrakis (3,5-di-tert-butyl-4-hydroxyhydrocinnamate) (AO1010), SA, and alcohol were purchased from Aladdin Chemical Co. All the reagents were directly used without further purification.

### 2.2. Preparation of PA6/BN Composites

PA6/BN composites were prepared by the melt-blending method. PA6 and BN were first vacuum-dried at 80 °C for 4 h and then mixed in a HAAKE mixer (Thermo Fisher, Waltham, MA, US) with a constant speed of 100 rpm at 225 °C for 10 min. To investigate the influence of SA on the composites, the proportion of PA6 and BN was fixed as 50:50. According to the content of SA, the obtained composites were named PA6/BN, PA6/BN/SA1, PA6/BN/SA2, PA6/BN/SA3, PA6/BN/SA4, PA6/BN/SA5, and PA6/BN/SA6. A dosage of 0.1 phr of Antioxidant 1010 is employed in each composite formula to effectively prevent thermal oxidative degradation during processing.

### 2.3. Characterization

The morphologies of the composites were observed by field emission scanning electron microscopy (FESEM, Nova NanoSEM 450, FEI, USA) at an accelerating voltage of 5 kV. All samples were frozen in liquid nitrogen and then fractured to obtain flat surfaces. The cross sections were sprayed with gold, and the thickness was about 5 nm.

The samples were subjected to X-ray diffraction (XRD) analysis on an X-ray diffractometer (D8 Advance, Bruker, Germany) employing a Cu-*K*_α_ radiation source. XRD data were collected from 10° to 80° at a scanning rate of 3°/min.

A differential scanning calorimeter (DSC; 200F3, Netzsch, Germany) characterized the melting and crystallization behaviors of the composites. First, the sample was heated from 50 °C to 250 °C at 50 °C/min, kept at 250 °C for 3 min to eliminate the thermal history, then cooled down to 50 °C at 10 °C/min, and finally heated to 250 °C again at 10 °C/min. The cooling and secondary heating were recorded for thermal performance analysis. The crystallinity (*X*_c_) of PA6 in the composites was calculated according to the following equation:(1)$$X_{c}=\frac{\Delta H_{m}}{\omega \cdot \Delta H_{m}^{0}}\times 100\%$$
where $\Delta H_{m}$ is the melting enthalpy of the sample, ω is the weight fraction of PA6 in the composite, and $\Delta H_{m}^{0}$ is the theoretically 100% melting enthalpy of PA6, 190 J g^−1^.

The thermal conductivity of the composite was measured at 25 °C with LFA 467 Nanoflash (Netzsch, Germany). The sample was a circular sheet with a diameter of 12.7 mm and a thickness of 0.9 mm. In addition, the sample was fixed between the bottom of an LED lamp and a radiator as a thermal interface material (TIM) to promote the heat dissipation of the LED lamp, and the surface temperature of the LED lamp was recorded by an infrared thermal imager (Fotric 365C, Fotric, Dallas, TX, USA).

The tensile properties of the samples were tested by a universal tensile tester (AGS-X, Shenzhen, China), and the tensile dumbbell-shaped samples were prepared by a Haake microinjection molding machine (Thermo Fisher, Waltham, MA, US) according to ISO 527-2-5A. The sample was stretched at a rate of 5 mm/min.

## 3. Results and Discussion

### 3.1. Morphology

As shown in the FESEM image of the composite in Figure 1, BN fillers in the composite without SA present aggregation, thicker lamellae, smooth surface, and obvious interface with the PA6 matrix, indicating that the interfacial adhesion between BN and PA6 is poor. With an increase in the SA content, BN fillers exhibit thinner and even dispersion in the matrix, indicating that the fillers are exfoliated, and the interfacial adhesion between the two phases becomes better. However, as the amount of SA is more than 5 phr, the interfacial adhesion of the composite is worse than that of the PA6/BN/SA5 composite.

In order to explore further the dispersion of SA in the composite, the cross section of the sample was etched with hot ethanol. The composites containing SA leave voids after being etched with hot ethanol, as shown in Figure 2. However, the morphology of the PA6/BN composite without SA presents no voids after being etched, indicating that only SA in the composite can be etched by hot ethanol. SA has some affinity with PA6 due to its polarity and structural similarity. SA molecules are typically composed of long-chain fatty acid molecules, which possess carboxylic acid groups that give SA a certain degree of polarity. Similarly, PA6 molecules also contain carboxylic acid and amide functional groups, which allow SA to interact with PA6 molecules through van der Waals forces and affinity interactions. These interactions help SA to disperse and dissolve in PA6. However, the affinity between SA and PA6 is not particularly strong. When the SA content is high, some SA molecules will be excluded to the interface between PA6 and BN, leading to the improvement of the compatibility between PA6 and BN. Therefore, SA was often used in PA6 composites to promote the compatibility between PA6 and fillers [44]. In this work, when the SA content is less than 3 phr, it is mainly distributed in the PA6 phase. When its content is at 3 phr, 4 phr, or 5 phr, due to the limited compatibility between SA and PA6, some SA is distributed at the interface between PA6 and BN, thus improving the interfacial adhesion between PA6 and BN. During the mixing process, the shear force acting on the matrix can be better transferred to the fillers, thus promoting the exfoliation and dispersion of BN fillers. However, when the content of SA is greater than 5 phr, the intermolecular interactions between SA molecules become stronger, eventually causing them to aggregate and form separate domains. As a result, the interface between PA6 and BN in the PA6/BN/SA6 composite becomes visible again.

In addition, it should be noted that in some of the samples depicted in Figure 1, the BN sheets were aligned in the direction of thermal conduction likely due to the shear force experienced during injection. To determine the orientation of the BN sheets in each sample, XRD analysis was performed. The XRD patterns for the samples can be found in Figure 3. It was observed that PA6 exhibited strong α-crystalline diffraction peaks at 2θ = 20.9° and 24.0°. As shown in Figure 3b, as the SA content increases, the α-crystalline diffraction peak of PA6 in the composite material slightly strengthens. However, in order to investigate the orientation of BN, the focus was placed on the diffraction peaks of BN within the composites. The sharp diffraction peaks at 2θ = 26.8° and 41.6° correspond to (002) and (100) crystal planes of BN, respectively [45]. The ratio of the intensity of (002) and (100) planes, which is recorded in Table 1, can be used to reflect the orientation of BN fillers in the samples. For the composites with an SA content of 3 phr, 4 phr, or 5 phr, the value of *I*_002_/*I*_100_ is significantly smaller compared with that for the other composites. This is consistent with the results observed by SEM. Due to the excellent dispersion of BN sheets in the composites of PA6/BN/SA3, PA6/BN/SA4, and PA6/BN/SA5, more BN sheets are oriented in the shear field.

### 3.2. Melting and Crystallization Behavior

The thermal conducting mechanism of PA6/BN composites is mainly phonon transmission. The transmission efficiency of phonons in the crystalline region is higher than that in the amorphous region. Therefore, the crystallinity of the polymer matrix in the composite has an important impact on phonon transmission. Figure 4 shows the DSC curves of cooling and secondary heating of the composites, and the relevant data are listed in Table 2. With the addition of SA, the melting temperature (*T*_m_) and crystallization temperature (*T*_c_) of PA6 in the composite decrease, which is mainly due to the plasticizing effect of SA in the PA6 matrix. Notably, the crystallinity of PA6 significantly increases with the increase in the SA content, which is consistent with the analysis of the XRD patterns. It is because the addition of SA promotes the exfoliation and dispersion of BN sheets, which increases the number of heterogeneous nucleation points in the composite, thus improving the crystallization of PA6. When the SA content is 5 phr, the crystallinity of PA6 in the composite is the highest, reaching 40.0%, which is 39.9% higher than that of pure PA6. As aforementioned, the dispersion of BN fillers decreases when the SA content is further increased, so the crystallinity of PA6 in the corresponding composite slightly decreases.

### 3.3. Thermal Conductivity

The thermal conductivities of the composites are shown in Figure 5a. When BN is added, the thermal conductivity of the composite is significantly higher than that of pure PA6, 0.3 W m^−1^ K^−1^. With the increase in the SA content, the thermal conductivity of the composite increases first and then decreases. The thermal conductivity of the PA6/BN/SA5 composite is the highest, 3.59 W m^−1^ K^−1^, which is 12 times and 2.6 times that of pure PA6 and the PA6/BN composite, respectively. It is consistent with the previous analysis. When the content of SA is 3 phr, 4 phr, or 5 phr, SA is mainly dispersed at the interface between BN sheets and the matrix, which improves their compatibility and reduces the scattering of phonons at the interface. Moreover, BN sheets in these composites exhibit excellent orientation, making it easy to construct thermally conductive paths. When the SA content is 5 phr, the crystallinity of the PA6 matrix in the PA6/BN/SA5 composite is the highest, which also contributes to the propagation of phonons in the matrix. Figure 5b exhibits the thermal conductivities of the composites prepared by the melt-blending method reported in the literature [7,37,38,39,40,41,42,43]. The data show that the thermal conductivity of the PA6/BN/SA5 composite is higher than that of other composites, demonstrating that adding a small amount of SA to PA6/BN composites is an effective strategy to fabricate composites with high thermal conductivity.

### 3.4. Transient Temperature Responses

The samples were used as the TIM to detect its heat dissipation effect on the LED lamp and to investigate its thermally conductive property. Figure 6 and Figure 7 show the thermal images and temperature–time profiles of the LED lamps using different composites as TIMs, respectively. After 120 s of power-on, the surface temperatures of the LED lamp with the PA6/BN, PA6/BN/SA1, PA6/BN/SA2, PA6/BN/SA3, PA6/BN/SA4, PA6/BN/SA5, and PA6/BN/SA6 composites as TIMs are 40.5 °C, 40.0 °C, 39.6 °C, 37.2 °C, 36.8 °C, 36.0 °C, and 38.8 °C, respectively, which are much lower than that of the LED lamp without TIM, 55.9 °C. The surface temperature of the LED lamp corresponding to the PA6/BN/SA5 composite is the lowest and quickly reaches the equilibrium point after 30 s.

### 3.5. Mechanical Property

The addition of inorganic fillers will have some effect on the mechanical properties of the composites. The tensile strength and Young’s modulus of pure PA6 were 66 MPa and 2500 MPa, respectively. The tensile properties of the composites are demonstrated in Table 3.

The tensile strength and Young’s modulus of the PA6/BN composite are 61.5 MPa and 2151 MPa, respectively, which are 6.8% and 14.0% lower than those of pure PA6. This suggests that the addition of a large amount of an inorganic filler decreases the strength and stiffness of PA6. As the amount of SA in the PA6/BN/SA composites increased, both the tensile strength and Young’s modulus showed a declining trend. At 5 phr, the composite exhibited a modest decrease in mechanical properties, with tensile strength and Young’s modulus dropping to 42.7 MPa and 1597 MPa, respectively. However, at 6 phr, the composite showed a significant decrease in mechanical properties, with tensile strength and Young’s modulus decreasing by 32% and 34%, respectively, compared with the PA6/BN/SA5 composite. This decrease can be attributed to the formation of separate phase domains of SA, which increases stress concentration and makes the composite more susceptible to fracture when subjected to external forces.

## 4. Conclusions

Adding thermal conductive fillers to polymers is an effective way to enhance their thermal conductivity. Among various preparation methods of thermal conductive composites, melt blending is the most convenient and practical method for large-scale production. In this study, PA6/BN composites were prepared using the melt-blending technique. To investigate the effect of small amounts of SA on the thermal conductivity of the composites, SA was added to the composite material.

When the content of SA is within a certain range, some SA can be uniformly distributed at the interface between BN sheets and the matrix. This not only enhances the compatibility of the materials but also promotes the exfoliation and dispersion of BN sheets. Furthermore, BN sheets are more likely to be oriented during the injection process. When the SA content is between 3 phr and 5 phr, the combined effect of better interface bonding, excellent orientation, and higher crystallinity of the matrix promotes the efficiency of phonon propagation. This leads to an increase in the thermal conductivity of the composite material. The highest thermal conductivity of the PA6/BN/SA5 composite material was achieved at 3.59 W m^−1^ K^−1^, which is 2.23 W m^−1^ K^−1^ higher than that of the PA6/BN composite. The resulting composites were used as TIMs (thermal interface materials) in heat dissipation experiments for LED lamps. The surface temperature of the LED lamp with the PA6/BN/SA5 composite as the TIM was the lowest. Although the tensile properties of the composite material decreased with the increase of the SA content, the decrease in the PA6/BN/5SA composite material was not significant. The tensile strength and Young’s modulus of the composite material were 42.7 MPa and 1597 MPa, respectively, which are suitable for various applications. This study provides a promising strategy for the preparation of composites with high thermal conductivity.

## Figures and Tables

**Figure 1 polymers-15-01887-f001:**
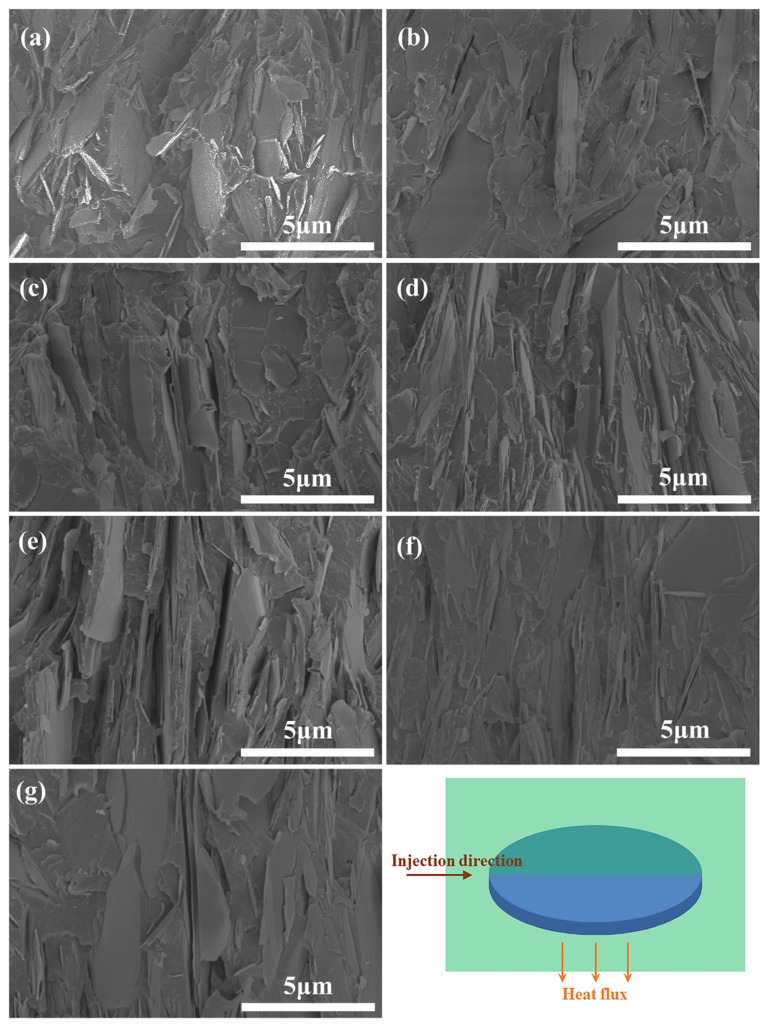
FESEM graphs of composites of (**a**) PA6/BN, (**b**) PA6/BN/SA1, (**c**) PA6/BN/SA2, (**d**) PA6/BN/SA3, (**e**) PA6/BN/SA4, (**f**) PA6/BN/SA5, and (**g**) PA6/BN/SA6.

**Figure 2 polymers-15-01887-f002:**
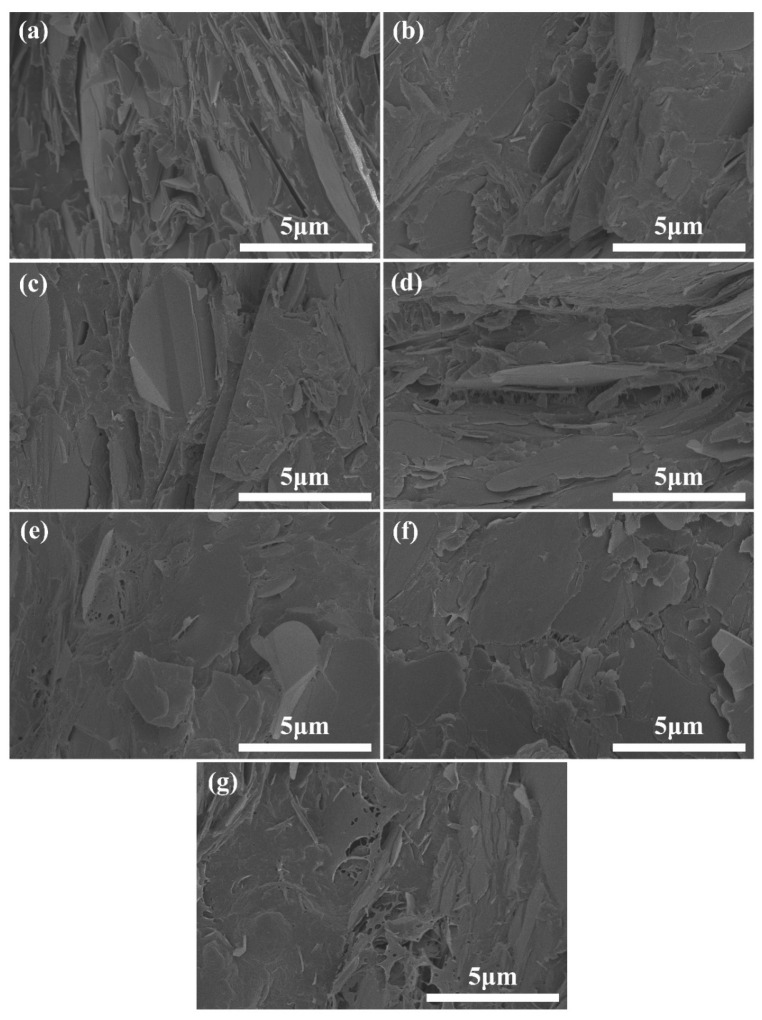
FESEM graphs of etched composites of (**a**) PA6/BN, (**b**)PA6/BN/SA1, (**c**) PA6/BN/SA2, (**d**) PA6/BN/SA3, (**e**) PA6/BN/SA4, (**f**) PA6/BN/SA5, and (**g**) PA6/BN/SA6.

**Figure 3 polymers-15-01887-f003:**
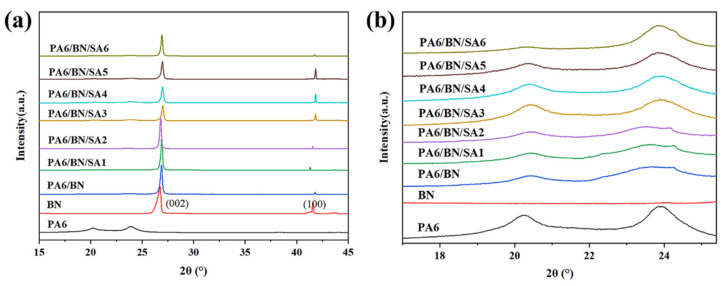
(**a**) XRD patterns of samples and (**b**) a zoomed area of the patterns.

**Figure 4 polymers-15-01887-f004:**
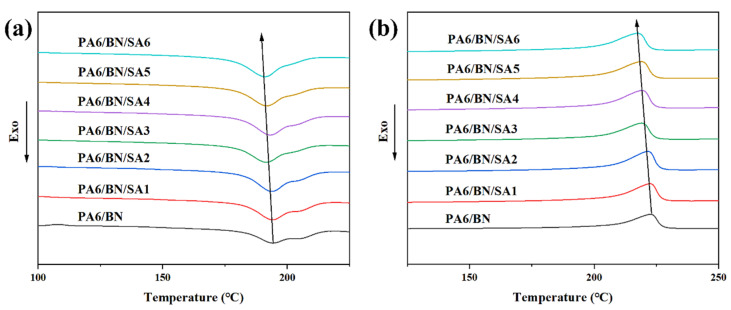
DSC curves of (**a**) cooling and (**b**) secondary heating of composites.

**Figure 5 polymers-15-01887-f005:**
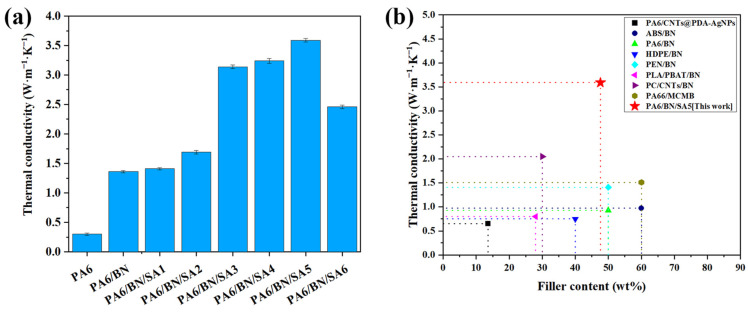
(**a**) Thermal conductivities of composites and (**b**) other composites prepared by melt-blending method [7,37,38,39,40,41,42,43].

**Figure 6 polymers-15-01887-f006:**
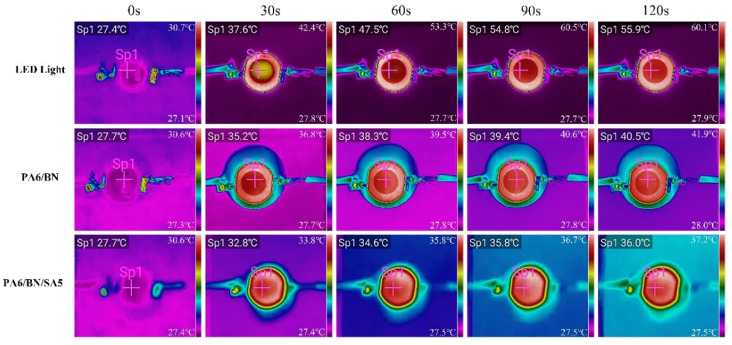
Thermal images of LED lamps.

**Figure 7 polymers-15-01887-f007:**
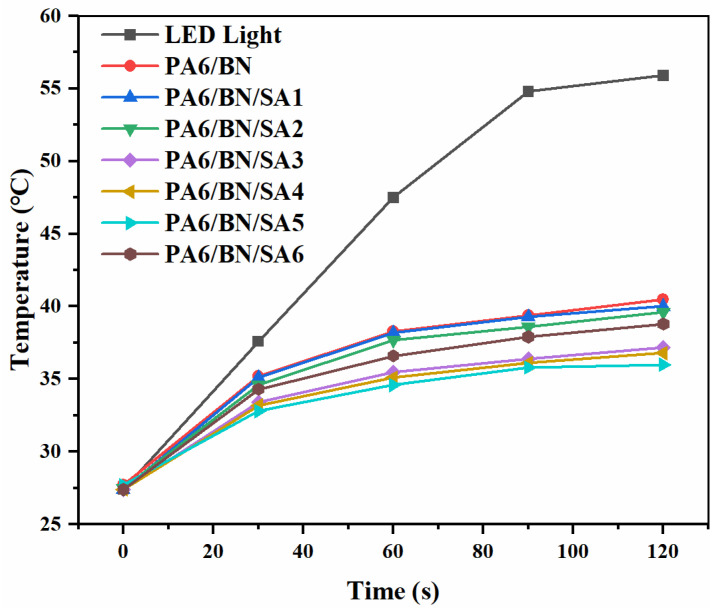
Temperature–time profiles of LED lamps.

**Table 1 polymers-15-01887-t001:** Ratio of intensity of (002) and (100) planes of BN in composites.

SA Content (phr)	0	1	2	3	4	5	6
*I*_002_/*I*_100_	256	181	139	9.78	9.68	5.98	102

**Table 2 polymers-15-01887-t002:** DSC data of composites.

Sample	*T*_m_ (°C)	Δ*H_m_* (J g^−1^*)*	*T*_c_ (°C)	Δ*H*_c_ (J g^−1^)	*X*_c_ (%)
PA6/BN	226.4	27.17	194.4	20.80	28.6
PA6/BN/SA1	222.4	31.40	194.1	31.64	33.4
PA6/BN/SA2	221.7	31.61	193.7	32.05	34.0
PA6/BN/SA3	219.0	33.16	191.6	26.64	36.0
PA6/BN/SA4	219.1	34.08	193.2	29.65	37.3
PA6/BN/SA5	218.9	36.17	191.7	29.05	40.0
PA6/BN/SA6	217.4	35.55	191.1	31.57	39.7

**Table 3 polymers-15-01887-t003:** Mechanical property of composites.

Sample	Tensile Strength (MPa)	Young’s Modulus (MPa)
PA6/BN	61.5 ± 0.5	2151 ± 32
PA6/BN/SA1	57.4 ± 1.8	2093 ± 30
PA6/BN/SA2	51.5 ± 0.6	1957 ± 26
PA6/BN/SA3	48.0 ± 1.6	1855 ± 24
PA6/BN/SA4	45.7 ± 0.9	1631 ± 27
PA6/BN/SA5	42.7 ± 0.8	1597 ± 26
PA6/BN/SA6	29.0 ± 1.0	1056 ± 29

## Data Availability

Not applicable.

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
