# Peer review of "Significant Improvement of Thermal Conductivity of Polyamide 6/Boron Nitride Composites by Adding a Small Amount of Stearic Acid"

_polymers, 2023, doi:10.3390/polym15081887_

Round 1

Reviewer 1 Report

I have some comments on the manuscript.

1) First of all, the manuscript is well prepared and may be of interest to many readers.

2) Line 27. Perhaps the tag "stearic acid" should be added to the keywords, since it is due to it that the authors increase the thermal conductivity of composites.

3) Line 89 and Table 1. What role does AO1010 play? There is no justification for adding it to the composition in the text.

4) The abbreviation "phr" should be deciphered.

5) Fig. 1 and 2 have the potential to give a lot of information, but they are very small and lack contrast. In this regard, the following two questions arise for them, trumpeting clarification.

6) Fig. 1, according to the authors, should demonstrate the effect of SA on improving BN exfoliation. There is a fairly obvious difference between all images and pic. f. But if you compare images a and d, then it is difficult to notice the difference.

7) In fig. 2, the authors note the areas from which SA was removed with ethanol. There are three questions here. 1) due to the quality of the images, the marked voids are very poorly distinguishable. 2) Why does the removal of SA occur not uniformly from the entire volume, but pointwise? Is there a bad mixing quality here? 3) The authors argue that only in the case of a composite with SA5, the addition of stearic acid is localized at the points of contact between PA6 and BN. But visually, on all images, the marks were made in the places of localization of the BN boundaries. Thus, the images in Fig. 1 and 2 need to be revised to better confirm the authors' conclusions.

8) Line 142. Is there any literature data on the compatibility of PA6 and SA, or is it just a hypothesis?

9) Line 152. There is an extra dot at the end of the fig. caption.

10) Section 3.3. The authors compare their results with the literature and show great success with the introduction of SA into composites. But it should be indicated what degree of filling the literary samples had in comparison with the composites of the authors. Because the possibility of thermal conductivity is ensured by the filler, and SA in this case plays an auxiliary role. Those. it is not entirely correct to compare literature data for only one parameter.

11) The bibliography section (and therefore the Introduction section) needs improvement. The absolute majority of references refer to authors from one country and ignore the experience of researchers from other countries. This looks biased and looks like self-citing.

Author Response

Reviewer 1

(1) First of all, the manuscript is well prepared and may be of interest to many readers.

R: Thank you.

(2) Line 27. Perhaps the tag "stearic acid" should be added to the keywords, since it is due to it that the authors increase the thermal conductivity of composites.

R: We've added "stearic acid" to the keywords. Thank you for your suggestions.

“Keywords: thermal conductivity; polyamide; boron nitride; stearic acid”

(3) Line 89 and Table 1. What role does AO1010 play? There is no justification for adding it to the composition in the text.

R: AO1010 is used as an antioxidant to inhibit the degradation of PA6 during processing. It has been mentioned in the revision. Usually, its dosage is around 0.1%. This study mainly focuses on the effect of stearic acid on the properties of composites, so the dosage of AO1010 remains unchanged.

“A dosage of 0.1 phr of Antioxidant 1010 is employed in each composite formula to effectively prevent thermal oxidative degradation during processing.”

(4) The abbreviation "phr" should be deciphered.

R: The abbreviation “phr” means Parts per hundred parts of resin, which is a commonly used unit in formulas. Therefore, if possible, there is no need to explain “phr”.

(5) Fig. 1 and 2 have the potential to give a lot of information, but they are very small and lack contrast. In this regard, the following two questions arise for them, trumpeting clarification.

Fig. 1, according to the authors, should demonstrate the effect of SA on improving BN exfoliation. There is a fairly obvious difference between all images and pic. f. But if you compare images a and d, then it is difficult to notice the difference.

In fig. 2, the authors note the areas from which SA was removed with ethanol. There are three questions here. 1) due to the quality of the images, the marked voids are very poorly distinguishable. 2) Why does the removal of SA occur not uniformly from the entire volume, but pointwise? Is there a bad mixing quality here? 3) The authors argue that only in the case of a composite with SA5, the addition of stearic acid is localized at the points of contact between PA6 and BN. But visually, on all images, the marks were made in the places of localization of the BN boundaries. Thus, the images in Fig. 1 and 2 need to be revised to better confirm the authors' conclusions.

R: Thank you for your suggestion. We have increased the contrast of Figure 1 and Figure 2. In addition, we carefully analyzed the distribution of SA in the composites and found that when the SA content was 3phr, 4phr or 5phr, some SA phases did distribute at the interface between PA6 and BN. Besides, some voids marked with red circles in Figure 2 only indicate that these etched areas are SA, not the distribution of all SAs. In the revision, the red circles have been removed.

“SA has some affinity with PA6 due to its polarity and structural similarity. SA molecules are typically composed of long-chain fatty acid molecules, which possess carboxylic acid and amine functional groups that give SA a certain degree of polarity. Similarly, PA6 molecules also contain carboxylic acid and amide functional groups, which allow SA to interact with PA6 molecules through van der Waals forces and affinity interactions. These interactions help SA to disperse and dissolve in PA6. However, the affinity be-tween SA and PA6 is not particularly strong. When the SA content is high, some SA molecules will be excluded to the interface between PA6 and BN, leading to the improvement of the compatibility between PA6 and BN. Therefore, SA was often used in PA6 composites to promote the compatibility between PA6 and fillers. [44] In this work, when the SA content is less than 3phr, it is mainly distributed in the PA6 phase. When its content is at 3phr, 4phr, or 5phr, due to the limited compatibility between SA and PA6, some SA is distributed at the interface between PA6 and BN, thus improving the interfacial adhesion between PA6 and BN. During the mixing process, the shear force acting on the matrix can be better transferred to the fillers, thus promoting the exfoliation and dispersion of BN fillers. However, when the content of SA is greater than 5 phr, the intermolecular interactions between SA molecules become stronger, eventually causing them to aggregate and form separate domains. As a result, the interface between PA6 and BN in the PA6/BN/SA6 composite becomes visible again.”

(6) Line 142. Is there any literature data on the compatibility of PA6 and SA, or is it just a hypothesis?

R: The compatibility of PA6 and SA has reported in other work. The reference has been cited in the revision.

“[44] Benkaddour A.; Demir E. C.; Jankovic N. C.; Kim C. I.; McDermott M. T.; Ayranci C. A hydrophobic coating on cellulose nanocrystals improves the mechanical properties of polyamide-6 nanocomposites. J. Compos. Mater. 2022, 56, 1775-1788.”

(7) Line 152. There is an extra dot at the end of the fig. caption.

R: Thank you for your carefulness. It has been revised.

“Figure 1. FESEM graphs of the composites of (a) PA6/BN, (b) PA6/BN/SA1, (c) PA6/BN/SA2, (d) PA6/BN/SA3, (e) PA6/BN/SA4, (f) PA6/BN/SA5, and (g) PA6/BN/SA6.”

(8) Section 3.3. The authors compare their results with the literature and show great success with the introduction of SA into composites. But it should be indicated what degree of filling the literary samples had in comparison with the composites of the authors. Because the possibility of thermal conductivity is ensured by the filler, and SA in this case plays an auxiliary role. Those. it is not entirely correct to compare literature data for only one parameter.

R: The content of thermal conductive fillers has a significant impact on the thermal conductivity of composites. In this study, the SA content is very low, but it can greatly improve the thermal conductivity of composites. By comparing with other works reported in the literature, it is more evident that this very small amount of auxiliary additive plays an important role in improving thermal conductivity.

(9) The bibliography section (and therefore the Introduction section) needs improvement. The absolute majority of references refer to authors from one country and ignore the experience of researchers from other countries. This looks biased and looks like self-citing.

R: Thanks for your suggestion. We have improved the Introduction section, and changed some references.

[3] Tomiak F.; Schneider K.; Schoeffel A.; Rathberger K.; Drummer Dietmar. Expandable Graphite as a Multifunctional Flame-Retarding Additive for Highly Filled Thermal Conductive Polymer Formulations. Polymers 2022, 14, 1613.

[13] Jang I.; Shin K. H.; Yang I.; Kim H.; Kim J.; Kim W. H.; Jeon S. W.; Kim J. P. Enhancement of thermal conductivity of BN/epoxy composite through surface modification with silane coupling agents. Colloid. Surface. A. 2017, 518, 64-72.

[29] Bragaglia M.; Lamastra F. R.; Russo P.; Vitiello L.; Rinaldi M.; Fabbrocino F.; Nanni F. A comparison of thermally conductive polyamide 6‐boron nitride composites produced via additive layer manufacturing and compression molding. Polym. Composite. 2021, 42, 2751-2765.

[33] Chung S. H.; Kim J. T.; Kim H.; Kim D. H.; Jeong S. W. Magnetic alignment of graphite platelets in polyimide matrix toward a flexible electronic substrate with enhanced thermal conductivity. Mater. Today Commun. 2022, 30, 103026.

Reviewer 2 Report

The results on thermal conductivity are excellent and worth publishing and fit well to Polymers journal, however the presentation of some results of characterisation need improvement: (1) Conclusions from SEM images are speculative, (2) XRD results are not preperly presented and evaluated Information on mechanical properies are completely missing. At such high content of the filler (50%) the mechanical properies would certainly be far from those of the PA6. If the authors cannot perform stress-strain measuremnts, they shoudl at least report how elastic/hard/brittle/bendable/fromable the materials feel when they are processed for thermal conductivity measurements and for TIM test.

The abstract does not contain two important pieces of information: that the PA6/BN blend had 50:50 ratio and that the highest thermal conductivity reached was 3.6 W/m/K.

The two most serious problems are:

1) SEM images (Fig. 1, Fig. 2) do not show any substantial differences between the materials, ecxept for that Fig. 2a and Fig. 2g differ from images Fig. 2b-2f. Fig. 2b-2f are virtually same (some voids present due to etching of stearic acid) but do not expain differences in thermal conductivities. Fig. 1a, 1d, 1f show preferential orientation of BN, Fig. 1b, 1c, 1g less orientation, Fig. 1e something between. But the data in Table 2 do not fit to the SEM: Accrodign to Table 2, Fig. 1d, 1e, 1f should show preferential orientation, Fig. 1a, 1b, 1c, 1g should show lower degree of oreintation.
Exfoliation will hardly be visible in SEM, TEM (transmission electron microscopy) would be necessary.

2) XRD data are not preoperly presented and evaluated:
Small peaks in Fig. 3a are hardly visible. Logarithmic scale shoudl be used, 2D plot with vertically shifted curves rather than 3D plot should be shown. Fig. 3b is not discussed at all. Crystallinity of the PA6 phase (2Theta between 20deg and 30deg) is not well visible in the graphs, is not discussed in the text and is not compared to the DSC results.

Furhter shortcomings:

It is not explained, what "AO1010/phr" in Table 1 means.

The sentence on lines 211 - 212 does not have a proper grammatical structure.

The corresponding author provided private email address, not institutional one.

Chapter 2.3, line 103: Sentence "All samples were frozen with liquid nitrogen to obtain smooth and flat surfaces." is nonsense. Smooth and flat surfaces cannot be reached by simple freezing. They can be reached by ultramicrotomy with a diamond knife. The SEM images look more like cryo-fractured surfaces. Cryo-fracture is good for inspecting the adhesion between filler and matrix, but less suitable for inspecting the distribution and dispersion of fillers.

Chapter 2.3, lines 105-106: Information on wavelength used is missing. It is not possible to calculate the scattering angles in XRD curves to scatterign vectors or lattice constants.

Anyway, the XRD curves (Fig. 3) shoudl have scattering vectors (in 1/nm or 1/A) on x-axis rather then scattering angle.

Table 1 contains 3 columns ("PA6/phr", "BN/phr", "AO1010/phr") that have the same value in whole column. Only values in column "SA/phr" changes from 1 to 6. This is why the table is useless - the tabular form does not contribute to better presentation of the composition, it is just waste of space. The same content can be communicated in 2 lines of text.

Author Response

Reviewer 2

The results on thermal conductivity are excellent and worth publishing and fit well to Polymers journal, however the presentation of some results of characterization need improvement: (1) Conclusions from SEM images are speculative, (2) XRD results are not properly presented and evaluated Information on mechanical properties are completely missing. At such high content of the filler (50%) the mechanical properties would certainly be far from those of the PA6. If the authors cannot perform stress-strain measurements, they should at least report how elastic/hard/brittle/bendable/formable the materials feel when they are processed for thermal conductivity measurements and for TIM test.

R: Thank you for your suggestion.

(1) For SEM images, the distribution of SA was observed through experiments. The impact of SA on the detachment and dispersion of BN was discussed in the revised manuscript, taking into account the published work.

“SA has some affinity with PA6 due to its polarity and structural similarity. SA molecules are typically composed of long-chain fatty acid molecules, which possess carboxylic acid and amine functional groups that give SA a certain degree of polarity. Similarly, PA6 molecules also contain carboxylic acid and amide functional groups, which allow SA to interact with PA6 molecules through van der Waals forces and affinity interactions. These interactions help SA to disperse and dissolve in PA6. However, the affinity be-tween SA and PA6 is not particularly strong. When the SA content is high, some SA molecules will be excluded to the interface between PA6 and BN, leading to the improvement of the compatibility between PA6 and BN. Therefore, SA was often used in PA6 composites to promote the compatibility between PA6 and fillers. [44] In this work, when the SA content is less than 3phr, it is mainly distributed in the PA6 phase. When its content is at 3phr, 4phr, or 5phr, due to the limited compatibility between SA and PA6, some SA is distributed at the interface between PA6 and BN, thus improving the interfacial adhesion between PA6 and BN. During the mixing process, the shear force acting on the matrix can be better transferred to the fillers, thus promoting the exfoliation and dispersion of BN fillers. However, when the content of SA is greater than 5 phr, the intermolecular interactions between SA molecules become stronger, eventually causing them to aggregate and form separate domains. As a result, the interface between PA6 and BN in the PA6/BN/SA6 composite becomes visible again.”

(2) The XRD patterns have been changed in the revision.

Figure 3. XRD patterns of PA6, BN and the composites.

(3) We have added the mechanical properties of the composites.

See the section of “3.5. Mechanical Property” in the revision.

The abstract does not contain two important pieces of information: that the PA6/BN blend had 50:50 ratio and that the highest thermal conductivity reached was 3.6 W/m/K.
R: The Abstract has been improved.

“This study investigates the effect of adding stearic acid (SA) on the thermal conductivity of Polyamide 6 (PA6)/boron nitride (BN) composites. The composites were prepared by melt blending, and the mass ratio of PA6 to BN was fixed at 50:50. The results show that when the SA content is less than 5 phr, some SA is distributed at the interface between BN sheets and PA6, which improves the interface adhesion of the two phases. This improves the force transfer from the matrix to the BN sheets, promoting the exfoliation and dispersion of BN sheets. However, when the SA content was greater than 5phr, SA tends to aggregate and form separate domains rather than being dispersed at the interface between PA6 and BN. Additionally, the well-dispersed BN sheets act as a heterogeneous nucleation agent, significantly improving the crystallinity of the PA6 matrix. The combination of good interface adhesion, excellent orientation, and high crystallinity of the matrix leads to efficient phonon propagation, resulting in a significant improvement in the thermal conductivity of the composite. The highest thermal conductivity of the composite is achieved when the SA content is 5 phr, which is 3.59 W·m-1·K-1. Moreover, the composite with 5 phr of SA used as a thermal interface material presents the best thermally conductive property. This study proposes a promising strategy for the preparation of composites with high thermal conductivity.”

The two most serious problems are:
1) SEM images (Fig. 1, Fig. 2) do not show any substantial differences between the materials, except for that Fig. 2a and Fig. 2g differ from images Fig. 2b-2f. Fig. 2b-2f are virtually same (some voids present due to etching of stearic acid) but do not explain differences in thermal conductivities. Fig. 1a, 1d, 1f show preferential orientation of BN, Fig. 1b, 1c, 1g less orientation, Fig. 1e something between. But the data in Table 2 do not fit to the SEM: According to Table 2, Fig. 1d, 1e, 1f should show preferential orientation, Fig. 1a, 1b, 1c, 1g should show lower degree of orientation.
Exfoliation will hardly be visible in SEM, TEM (transmission electron microscopy) would be necessary.

R: Figure 1 has been revised. The data in Table 2 could fit to the SEM. In addition, this section only describes the morphological characteristics, and its impact on thermal conductivity is introduced in the section of “3.3. Thermal conductivity”.

Besides, ultrathin cryo-sectioning is the traditional method for preparing TEM samples may be difficult for composite materials containing boron nitride due to their high hardness and uneven distribution. Although the hardness of boron nitride is slightly lower than that of diamond, it is still very high and close to the hardness of diamond. Therefore, when using a diamond blade to cut high hardness materials such as boron nitride, the diamond blade is prone to damage or wear. There is almost no TEM characterization of composites containing BN in the published literature. Therefore, if SEM graphs can demonstrate the dispersion of BN, can TEM not be performed?

2) XRD data are not properly presented and evaluated:
Small peaks in Fig. 3a are hardly visible. Logarithmic scale should be used, 2D plot with vertically shifted curves rather than 3D plot should be shown. Fig. 3b is not discussed at all. Crystallinity of the PA6 phase (2Theta between 20deg and 30deg) is not well visible in the graphs, is not discussed in the text and is not compared to the DSC results.
R: The XRD patterns have been revised. The diffraction peaks of PA6 can be visible in Figure 3b.

“It was observed that PA6 exhibited strong α-crystalline diffraction peaks at 2θ = 20.9° and 24.0°. As shown in Figure 3(b), as the SA content increases, the α-crystalline diffraction peak of PA6 in the composite material slightly strengthens.”

Further shortcomings:
It is not explained, what "AO1010/phr" in Table 1 means.
R: Table 1 has been removed in the revision.

The sentence on lines 211 - 212 does not have a proper grammatical structure.
R: It has been revised.

“The samples were used as the TIM to detect its heat dissipation effect on the LED lamp and to investigate its thermally conductive property.”

The corresponding author provided private email address, not institutional one.
R: It has been revised.

Chapter 2.3, line 103: Sentence "All samples were frozen with liquid nitrogen to obtain smooth and flat surfaces." is nonsense. Smooth and flat surfaces cannot be reached by simple freezing. They can be reached by ultramicrotomy with a diamond knife. The SEM images look more like cryo-fractured surfaces. Cryo-fracture is good for inspecting the adhesion between filler and matrix, but less suitable for inspecting the distribution and dispersion of fillers.

R: It has been revised.

“All samples were frozen in liquid nitrogen and then fractured to obtain flat surfaces.”

Chapter 2.3, lines 105-106: Information on wavelength used is missing. It is not possible to calculate the scattering angles in XRD curves to scattering vectors or lattice constants.

R: The radiation source has been provided.

“The samples were subjected to X-ray diffraction (XRD) analysis on an X-ray dif-fractometer (D8 Advance, Bruker, Germany) employing a Cu-Kα radiation source. XRD data were collected from 10° to 80° at a scanning rate of 3°/min.”

Anyway, the XRD curves (Fig. 3) should have scattering vectors (in 1/nm or 1/A) on x-axis rather than scattering angle.
R: The X-axis of an XRD (X-ray diffraction) pattern usually represents the diffraction angle of the crystal in the sample, given in degrees (°) or radians (rad).

Table 1 contains 3 columns ("PA6/phr", "BN/phr", "AO1010/phr") that have the same value in whole column. Only values in column "SA/phr" changes from 1 to 6. This is why the table is useless - the tabular form does not contribute to better presentation of the composition, it is just waste of space. The same content can be communicated in 2 lines of text.

R: Table 1 has been removed in the revision.

Reviewer 3 Report

The manuscript entitled “Significant improvement of thermal conductivity of polyamide 2 6/boron nitride composites by adding a small amount of stearic 3 acid” by Hui Fang et al has a number polymers-2292093-peer-review-v1. This is very good work and well written. However, few English mistakes are there in the manuscript, which are given in the attched file and submitted o editorial office. So the manuscript can be accepted in polymers after its minor revision.

Author Response

Reviewer 3

The manuscript entitled “Significant improvement of thermal conductivity of polyamide 2 6/boron nitride composites by adding a small amount of stearic acid” by Hui Fang et al has a number polymers-2292093-peer-review-v1. This is very good work and well written. However, few English mistakes are there in the manuscript, which are given below. So the manuscript can be accepted in polymers after its minor revision.

Needed corrections

Abstract

  1. All right

Keywords:

  1. All right
  2. Introduction
  3. Page 2, line 57. ---thermal conductivity[21].

A gap is necessary before the reference. So change as---thermal conductivity [21].

R: Thanks. It has been revised.

  1. Page 2, lines 75-78. Generally, as to the thermally-conductive composites prepared by the melt blending method, a large amount of BN fillers were usually added to significantly enhance the thermal conductivity.

Change as

Generally, when a thermally-conductive composites prepared by the melt blending method, a large amount of BN fillers were usually added to significantly enhance the thermal conductivity.

R: Thanks. It has been revised.

  1. Page 2, lines 79-82. Notably, compared with other methods, in most of the reported work, although much more BN fillers were applied in the melt blending process, the thermal conductivity of the composite was still less than 2.0 W·m-1·K-1, which was not enough in some applicants.

The above sentence is not proper. So change as

In most of the reported work with other methods, although much more BN fillers were applied in the melt blending process, the thermal conductivity of the composite was still less than 2.0 W·m-1·K-1, which was not enough in some applicants.

R: Thanks. It has been revised.

  1. Experimental

2.1 Materials

  1. All right.

2.2 Preparation of PA6/BN composites

  1. All right.

2.3. Characterization

  1. All right.
  2. Results and Discussion

3.1. Morphology

  1. Page 4, line 130. To further explore the dispersion of SA in the composite,---

Change as

In order to explore further the dispersion of SA in the composite,---

R: Thanks. It has been revised.

3.2. Melting and crystallization behavior

  1. All right.

3.3. Thermal conductivity

  1. Page 6, lines 199-201. Besides, when SA content is 5 phr, the crystallinity of the PA6 matrix in the PA6/BN/SA5 composite is the highest, which also contributes to the propagation of phonons in the matrix.

The word “Besides” is not necessary. So change as

When SA content is 5 phr, the crystallinity of the PA6 matrix in the PA6/BN/SA5 composite is the highest, which also contributes to the propagation of phonons in the matrix.

R: Thanks. It has been revised.

3.4. Transient temperature responses

  1. All right.
  2. Conclusions
  3. All right.

Acknowledgement

  1. Authors should add a sentence to acknowledge to their head of department or institute in addition to the given acknowledgement.

R: Thank you. We have mentioned it.

“We also appreciate Li Huang and Xingfang Huang from the College of Materials Science and Engineering in Fujian University of Technology for their help in partial measurement and characterization.”

References

  1. All right.

Tables

  1. All right.

Figures

  1. All right

Round 2

Reviewer 2 Report

The authors write: "SA molecules are typically composed of long-chain fatty acid molecules, which possess carboxylic acid and amine functional groups". Where exactly in the SA with chemical formula CH3-(CH2)16-COOH is the amine functional group located? I can see there no nitrogen.

The authors write: "PA6 molecules also contain carboxylic acid and amide functional groups". Where exactly in the PA6 with chemical formula [-CO-NH-(CH2)5-]n is the carboxylic acid (COOH) loacted? I can see there only carbonyl group (CO) or better to say amide group (-CONH-).

Author Response

(1) The authors write: "SA molecules are typically composed of long-chain fatty acid molecules, which possess carboxylic acid and amine functional groups". Where exactly in the SA with chemical formula CH3-(CH2)16-COOH is the amine functional group located? I can see there no nitrogen.

R: Thank you. There is indeed no amine group on SA, which is our mistake. It has been revised.

“SA molecules are typically composed of long-chain fatty acid molecules, which possess carboxylic acid groups that give SA a certain degree of polarity.”

(2) The authors write: "PA6 molecules also contain carboxylic acid and amide functional groups". Where exactly in the PA6 with chemical formula [-CO-NH-(CH2)5-]n is the carboxylic acid (COOH) loacted? I can see there only carbonyl group (CO) or better to say amide group (-CONH-).

R: Thank you. The chemical formula of PA6 is [-CO-NH-(CH2)5-]n. However, the end groups of the molecular chain of PA6 contain carboxylic acid groups.